Validation of a battery of inhibitory control tasks reveals a multifaceted structure in non-human primates

Loyant Louise 1 louise.loyant@port.ac.uk
Waller Bridget M. 2
http://orcid.org/0000-0002-4480-6781 Micheletta Jérôme 1
http://orcid.org/0000-0002-0784-5167 Joly Marine 1
1 Centre for Comparative and Evolutionary Psychology, Department of Psychology, University of Portsmouth , Portsmouth, Hampshire , United Kingdom
2 Department of Psychology, Nottingham Trent University , Nottingham , United Kingdom
Barrett Louise
Electronic publication date: 2022 Feb 9
Publication date: 2022
Volume: 10
Electronic Location ID: e12863
Received 2021 Sep 28; Accepted 2022 Jan 9
Copyright: © 2022 Loyant et al.
Copyright year: 2022
Copyright holder: Loyant et al.
License: This is an open access article distributed under the terms of the Creative Commons Attribution License, which permits unrestricted use, distribution, reproduction and adaptation in any medium and for any purpose provided that it is properly attributed. For attribution, the original author(s), title, publication source (PeerJ) and either DOI or URL of the article must be cited.
License URL: https://creativecommons.org/licenses/by/4.0/

Keywords: Macaque, Inhibitory control, Go/No-go task, Reversal learning task, Distraction task, Non-human primates, Battery of tasks, Validity, Executive function, Self-control

Funding: University of Portsmouth Faculty Bursary International Primatological research grant Primate Society of Great Britain research grant This study was supported by the University of Portsmouth Faculty Bursary, the International Primatological research grant and the Primate Society of Great Britain research grant. There was no additional external funding received for this study. The funders had no role in study design, data collection and analysis, decision to publish, or preparation of the manuscript.

==============================
Inhibitory control, the ability to override an inappropriate prepotent response, is crucial in many aspects of everyday life. However, the various paradigms designed to measure inhibitory control often suffer from a lack of systematic validation and have yielded mixed results. Thus the nature of this ability remains unclear, is it a general construct or a family of distinct sub-components? Therefore, the aim of this study was first to demonstrate the content validity and the temporal repeatability of a battery of inhibitory control tasks. Then we wanted to assess the contextual consistency of performances between these tasks to better understand the structure of inhibitory control. We tested 21 rhesus macaques (Macaca mulatta, 12 males, nine females) in a battery of touchscreen tasks assessing three main components of inhibitory control: inhibition of a distraction (using a Distraction task), inhibition of an impulsive action (using a Go/No-go task) and inhibition of a cognitive set (using a Reversal learning task). All tasks were reliable and effective at measuring the inhibition of a prepotent response. However, while there was consistency of performance between the inhibition of a distraction and the inhibition of an action, representing a response-driven basic form of inhibition, this was not found for the inhibition of a cognitive set. We argue that the inhibition of a cognitive set is a more cognitively demanding form of inhibition. This study gives a new insight in the multifaceted structure of inhibitory control and highlights the importance of a systematic validation of cognitive tasks in animal cognition.

Introduction

Living in a complex social environment requires animals to manage their impulsive behaviours to maintain group cohesion and survival. For example, in the presence of a higher-ranking conspecific, an individual needs to inhibit aggressive behaviours when competing over food (Amici, Aureli & Call, 2008; Byrne & Bates, 2007) or over a mating partner (Estep et al., 1988; Lindsay et al., 1976). These inhibitory processes constitute core components of executive functions; a family of top-down cognitive control processes which support goal directed behaviours (Aron, 2007; Banich, 2009; Diamond, 2013; Duque & Stevens, 2017; Miyake et al., 2000; Nigg, 2017). Here we define inhibitory control as the ability to deliberately control a reflexive, automatic, or pre-learned response and therefore achieve a more complex goal (Dempster & Corkill, 1999; Dillon & Pizzagalli, 2007; Friedman & Miyake, 2017; Miller et al., 2019; Miyake et al., 2000). A strong internal predisposition or an external distractor, tempting but counterproductive, irrelevant to the individual’s goal, must be overridden in order to do what is more appropriate or needed (Diamond, 2013; Dillon & Pizzagalli, 2007; Nigg, 2017). To cover the main domains of inhibitory control, we focus particularly on three of the most commonly described inhibitory processes in the literature: distraction inhibition (e.g., control of an emotional response to an internal or external distractor, in order to focus on a goal), action inhibition (e.g., inhibition of a prepotent, unwanted, reflexive motoric action) and cognitive set inhibition (e.g., inhibition of a pre-learned cognitive set to flexibly adjust behaviours; see Aron, 2007; Diamond, 2013; Dillon & Pizzagalli, 2007; Nigg, 2017).

An important research question that has been a source of controversy in both neuropsychological and cognitive studies is to what extent these inhibitory processes can be considered unitary in the sense that they are reflections of the same underlying mechanism or ability (Dempster & Corkill, 1999; MacLeod, 2007; Miyake et al., 2000; Nigg, 2017). For instance, to support this unitary hypothesis, Duckworth & Kern (2011), in a large-scale meta-analysis (based on over 33,000 adult participants), demonstrated a moderate but significant convergence of several inhibition related measures such as the Go/No-go task (the subjects need to respond to Go stimuli while inhibiting response to an unrewarded No-go stimuli) or the Stroop task (the subjects need to inhibit an interference from a distractor while doing a task). Similarly, in the animal cognition field, MacLean et al. (2017) found that the cylinder task (a common inhibitory control task in which the animal needs to inhibit reaching directly for food through the transparent surface of a cylinder) and two detour tasks (the subject needs to circumvent an obstacle to get a reward) loaded onto the same factor in a large-scale battery of cognitive tasks in 552 dogs (Canis familiaris). MacLean et al. (2014) also found a strong correlation in performances between the A-not-B task (the subject must inhibit a previously rewarded behaviour to learn a new reward-contingency) and the cylinder task across 23 primate species.

In contrast, several authors have proposed that inhibition-related processes are instead a family of functions rather than a single unitary construct (Friedman & Miyake, 2004, 2017; Nigg, 2000, 2017). In humans, Friedman & Miyake (2004), using common inhibitory control tasks, found that two inhibitory factors: “prepotent response inhibition” and “resistance to distractor interference” were closely related, but both were unrelated to “resistance to proactive interference”, a form of cognitive set inhibition. In the animal cognition literature, several studies tested dogs (C. familiaris) in common inhibitory control tasks, including the Cylinder and the A-not-B. The authors found that the dogs’ performances in these tasks were not correlated (Bray, MacLean & Hare, 2014; Brucks et al., 2017; Fagnani et al., 2016; Vernouillet et al., 2018). A similar result was also found in a study comparing wolves (Canis lupus) and dogs’ performances in a cylinder task and in a detour paradigm (Marshall-Pescini, Virányi & Range, 2015). From these results, the authors concluded that inhibitory control would be context specific and of a diverse structure (Bray, MacLean & Hare, 2014; Brucks et al., 2017; Fagnani et al., 2016; Vernouillet et al., 2018).

However these mixed results and lack of correlation are difficult to interpret as evidence for separable inhibition-related processes for several reasons (Friedman & Miyake, 2004; Miyake et al., 2000). The first reason is that researchers often use these tasks and assume that they measure inhibitory control but without providing any justification of the choice of the task (Friedman & Miyake, 2004). Thus the content validity (defined as the degree to which a measurement is representative of the targeted construct, Haynes, Richard & Kubany, 1995) is rarely evaluated (Friedman & Miyake, 2004). According to Völter et al. (2018), to assess the content validity of a task, researchers should agree on features defining the ability of interest and look at characteristic responses’ patterns. For instance, in inhibitory control tasks authors should demonstrate that a prepotent response (dominant and automatic response to a stimulus) have been triggered by the test conditions (Völter et al., 2018).

The second complication arises from the fact that common inhibitory control tasks tend to suffer poor repeatability, i.e., multiple exposures to the same task often lead to inconsistent individual’s performance over time (Friedman & Miyake, 2004; Völter et al., 2018). Yet this test-retest reliability is necessary before considering the validity of a task (Biro & Stamps, 2015; Griffin, Guillette & Healy, 2015; Völter et al., 2018). In the animal cognition literature, the repeatability of inhibitory control measurements is rarely assessed, and the results are mixed. For instance, great tits (Parus major) demonstrated a significant repeatability of performances in successive reversal learning tasks (Cauchoix et al., 2017). Similarly, Australian magpies (Cracticus tibicen) tested as juveniles in the cylinder task and reversal learning task repeated their performance as adults (Ashton et al., 2018). However, the performance of robins (Petroica longipe) in the cylinder task did not seem repeatable over a year (Shaw, 2017). This lack of valid and reliable cognitive measurements, often referred as the “replicability crisis”, is unfortunately a common issue in psychology experiments (for review see Lindsay, 2015).

A third difficulty is the task impurity problem, i.e., no tasks are pure measurement of a single cognitive process. Inhibitory control tasks often involve other cognitive (e.g., memory) or non-cognitive processes (e.g., personality) that are not directly relevant to the targeted function (Friedman & Miyake, 2004; Gärtner & Strobel, 2019; Miyake et al., 2000; Völter et al., 2018). For instance, the cylinder tasks, one of the benchmark tests in large inter-species comparisons (MacLean et al., 2014, 2017), is the subject of vivid debates (for review see Kabadayi, Bobrowicz & Osvath, 2018; Shaw & Schmelz, 2017). Factors such as prior experience (Duque & Stevens, 2017; Kabadayi, Bobrowicz & Osvath, 2018; van Horik et al., 2019; Vernouillet et al., 2018), can dramatically influence a subject’s performance on this detour task. Various authors advocate to circumvent this task impurity problem by using a battery of tasks putatively measuring the same ability (but differing in other task demands) to reveal a common underlying cognitive construct (Cauchoix et al., 2018; Friedman & Miyake, 2017; Primates et al., 2019; Shaw & Schmelz, 2017; Völter et al., 2018). If inhibitory control was a common ability, multiple tasks putatively measuring the same inhibitory process, should demonstrate cross-contextual consistency in individuals’ inhibitory performance (Shaw & Schmelz, 2017; Völter et al., 2018).

Therefore, the aim of this study was to validate a battery of inhibitory control tasks in non-human primates: (1) by assessing the content validity of three tasks covering the main domains of this inhibitory ability, (2) by demonstrating the necessary temporal repeatability of these tasks. Finally, we wanted to investigate the structure of inhibitory control by looking at the cross-contextual consistency between tasks. The goal here was to assess whether the main components of inhibitory control fall under the same common inhibitory ability or whether they are part of a family of distinct sub-components.

Rhesus macaques (Macaca mulatta), phylogenetically close to human species, represent an interesting model for our battery of inhibitory control tasks. This species, exhibits complex social organisation (Thierry, 2000), possesses enhanced general intelligence (Reader & Laland, 2002) and can perform computerized testing (e.g., Gazes et al., 2013; Washburn, 1994; Washburn, Harper & Rumbaugh, 1994). The Macaca genus has been tested in several inhibitory control tasks, for instance researchers have demonstrated the Stroop effect (interference effect of an incongruent stimulus, Lauwereyns et al., 2000) and the emotional Stroop effect in Japanese macaques (Macaca fuscata; Hopper et al., 2021). Barbary Macaques (Macaca sylvanus) have been tested in a modified version of the cylinder task and in cognitive flexibility tasks (Rathke & Fischer, 2020). Rhesus macaques have also been tested in reversal learning tasks (Rayburn-Reeves, James & Beran, 2017) and in a stop task associated with emotional pictures (Vardanjani et al., 2021). We developed a battery of touchscreen tasks of inhibitory control from well-established tasks used in human and animal research, and which have been tested for validity and reliability (Rana & Rao, 2013; Thomas, Rao & Devi, 2016; Wöstmann et al., 2013). The touchscreen technology is an extremely flexible tool which allows accurate record of subject’s answer while controlling for important confounding factors (Kangas & Bergman, 2017). We focused on the major components of inhibitory control. To investigate the inhibition of a distraction, we conducted a Distraction task. In this task, a subject must inhibit a dominant and prepotent emotional response to a distractor (Allritz, Call & Borkenau, 2016; Bethell et al., 2016; Isaac et al., 2012; Stroop, 1935). To investigate the inhibition of an action, we conducted a Go/No-go task. Here a subject learns to develop a prepotent motor response to frequently appearing target and must withhold it to less frequently appearing non target (Aron, 2007; Diamond, 2013; Dillon & Pizzagalli, 2007; Duckworth & Kern, 2011). Lastly, to assess inhibition of a cognitive set, we conducted a Reversal learning task. In this task, a subject must inhibit a pre-learned rule to adopt a new set of rules (Bray, MacLean & Hare, 2014; Jelbert, Taylor & Gray, 2016).

We first expected to demonstrate the content validity of our three inhibitory control tasks, i.e., a prepotent response (dominant and automatic response to a stimulus) was generated by the test conditions. We further expected to demonstrate temporal repeatability of the tasks by showing that the rank of the individual’s performances within the group of subjects and within the same task, would be repeatable over two time points (2 weeks apart). On one hand, we did not want this interval to be too long to avoid dramatic changes in the internal and external states of the subjects (Bell, Hankison & Laskowski, 2009; Shaw & Schmelz, 2017). On the other hand, we did not want this interval to be so short that there would be an influence of carry over effects (Bell, Hankison & Laskowski, 2009). Finally, we expected that these three tasks would not demonstrate cross-contextual repeatability of the individuals’ performances thus supporting the theory of the multifaceted structure of inhibitory control.

Materials and Methods

Data were collected as previously described in Loyant et al. (2021).

Subjects

All the adult rhesus macaques (M. mulatta) taking part in this study were from the breeding colony of the Medical Research Council’s Center for Macaques (MRC-CFM) in Porton Down, United Kingdom. Each group had access to an indoor free-roaming room (3.35 × 8.04 × 2.8 m) and an adjacent caged area (1.5 × 6.12 × 2.8 m), with a minimum total space of 3.5 m3/breeding animal in the largest groups. All rooms were temperature controlled (20 °C ± 5) with humidity at 55% +/−10. Each free-roaming area had a large bay window at one end facing outdoors and allowing a natural day-night cycle. At the other end of each room was an internal window fitted with movable mirrors which the monkey could control using a handle, allowing them to view the activities outside their area. Rooms were enriched with climbing structures (platforms, poles, fire hose and ladders) and enrichment devices (food puzzles, boxes, plastic barrels and balls, and small plastic blocks attached to structures or walls). Subjects received a supply of fruit and vegetables, dried forage mix (cereal, peas, beans, lentils etc.), bread and boiled eggs, in the morning and afternoon, with enough food to last for a 24 h period. All subjects had access to water and food prior to and during the experiment. Eighteen of the subjects already participated in a behavioural study involving looking at pictures (Bethell et al., 2019) and all of them were familiar with basic training and clicker procedures. However, none of them had experience with touchscreen experiments. Thirty subjects (14 males, 16 females; aged from 3 to 17 years old, mean age in years M ± SD = 8.10 ± 4.05, N = 30) started the touch screen training phases (see Supplemental materials, Training phases) but only 21 (12 males and nine females, aged from 3 to 17 years old, mean age in years M ± SD = 8.9 ± 4.41) successfully completed the training and were able to take took part in the experiment. The subjects were housed in 14 different social groups with an average of 12 individuals per group. Hierarchy, calculated in each group using David’s scores (David, 1987), was provided by the head of research of the facility (see Supplemental materials, Rank calculations). Agonistic behaviours including threats, displacements, chases, and physical conflict were recorded to assess the hierarchy. The caretakers regularly monitored the groups, and David’s scores were updated accordingly. Using video recordings of training and test sessions, a blind observer coded agonistic interactions between the tested individual and other conspecifics to verify the given ranks for each group (see Supplemental material). If a male never lost, he was considered high ranking. If the female never lost against other females, she was considered high-ranking, otherwise she was considered low-ranking.

Ethics

This study was approved by the Animal Welfare and Ethical Review Body of the University of Portsmouth, AWERB no. 4015B and by the MRC-CFM’s Animal Welfare and Ethical Review Body, ARWEB no. CFM2019E002 and was part of the Macaque Cognition Project. All research was carried out in accordance with ethical guidelines for work with non-human primates (NC3Rs, 2017).

Apparatus

To minimise stress, all tests were conducted in the enclosure, with no isolation from the social group; meeting recent and important ethical considerations on animal welfare and cognition (Cronin et al., 2017; Jacobson et al., 2018). The macaques had free and voluntary access to the apparatus and were never restrained; at any point subjects could leave the experiment and return voluntarily. For the experimental task, the set up was customized to be transported from one cage to another. Outside the cage, a laptop was connected via USB and HDMI cables to a capacitive touchscreen (ELO 1590L, frequency of 60 Hz, 19″ in diagonal, resolution 1,280 × 1,024 pixels). The program Elo touch solution 6.9.20 was used for calibration. The laptop screen duplicated the touchscreen display to be able to follow the experiment’s progress. The touchscreen was attached to the cage bars and the position was adjusted to each individual. All experimental procedures including stimulus presentation and response collection were carried out using MATLAB (version R 2018b, using Psychtoolbox-3.0.15 functions), under Windows 10. The MATLAB scripts were specifically conceived for the needs of this study; an individual progression file allowed the experimenter to abort and come back to the same point of a running session (see Supplemental Material for MATLAB CODE). If a trial was aborted the response latency was not recorded. The computer gave auditory feedback in response to the subject’s performance. All sessions were videotaped with one digital video camera (Sony HDR-CX330EB). The rewards (dry raisins) for each correct answer were given by hand.

General procedure

When the program for a specific task was launched, the experimenter entered the name of the individual. When more than one individual per cage were tested or when other individuals from the group were interacting with the touch screen, a research assistant was distracting the other macaques at the opposite side of the room. Every session was initiated by the subject touching a red cross located in the centre of the screen, starting the time recording. The session was aborted when another individual displaced the tested individual or interacted with the touchscreen. If the subject left the testing area or was not focusing attention on the screen the session was aborted. If the target was not touched within the specified time limit (see task descriptions below for specific time limit), the built-in timer of the program was paused, and a red cross appeared in the centre of the screen until the session was resumed by touching it. The response latencies above the time limits were not used in the analysis. If the subject stayed inactive for more than 5 min the experiment was stopped and continued the next testing day, if the subject did not participate for three testing days in a row the subject was excluded from the task.

Inhibitory control tasks

For the task battery we chose three tasks covering the main domains of inhibitory control: inhibition of distraction, inhibition of action and inhibition of a cognitive set (see Fig. 1). The tasks were conducted in the same order as they were built upon the previous task (see Fig. 1 for a visual presentation of the tasks and timeline): first a Distraction task (a target touching task with pictorial stimuli as distractors). This task was repeated (to assess temporal repeatability), for each subject, 2 weeks apart. Once this task was completed for the second time, the subjects were tested (the next testing day) on the Go/No-go task (a novel unrewarded stimulus was introduced). Once, the subjects were tested a second time on the Go/No-go task, they were tested on the reversal learning task (built upon the previously rewarded and unrewarded stimuli). Finally they finished the experiment by being tested a second time on the Reversal learning task 2 weeks after. As in previous batteries of tasks in animals (Beran & Hopkins, 2018; Herrmann et al., 2007, 2010; Lacreuse et al., 2014; MacLean et al., 2017; Wobber et al., 2014), the order of tasks was the same for all subjects. Although this design cannot eliminate the possibility of order effects (i.e., the participation on a given task affects performance on subsequent measures), it ensures consistency across subjects, permitting direct comparisons of performances across time and tasks. Besides, we wanted our subject to have the same experience with inhibitory control testing as this ability can be learned (Diamond, 2013) and is directly influenced by previous inhibitory control testing (Radel, Gruet & Barzykowski, 2019).

Figure 1 Schematic representation of the touchscreen tasks procedures and aims of the study.

The Distraction task (inhibition of a distraction), the Go/No-go task (inhibition of an action) and the Reversal learning task (inhibition of a cognitive set) are presented. The aims of the study are also presented: 1. Content validity, 2. Temporal repeatability (with timeline) and 3. Contextual repeatability.

Inhibition of distraction: distraction task

The Distraction task we used is a variant of the Emotional Stroop task in which human participants name the colours of words that differ in emotional valence, with a longer response latency to negative words (Bar-Haim et al., 2007; Williams, Mathews & MacLeod, 1996). We used a simplified version of the Emotional Stroop task (Allritz, Call & Borkenau, 2016, in chimpanzees, Pan troglodytes) in which a prepotent response to an emotional stimulus interferes with the goal of the task.

Design

Every session was initiated by the subject touching a red cross in the centre of the screen. Then the timer started, and the subject had to touch a target (a red rectangle of 10 × 13 cm) randomly displayed at the far left or right of the screen. When the subject successfully touched the target, a high-pitched chime was played, the timer was stopped, and the reward was given. After an inter-trial of 2,000 ms with only a white background displayed, the next trial was presented. Such a trial without a distractor was considered as a “Control” trial. Two “Control” trials were followed by a block of four trials with pictures from the same categories (either four pictures of objects, neutral or threatening conspecific faces, see Fig. 1 and online supplemental materials for MATLAB codes and stimuli). Each block of pictures of the same category was seen 2 times per session. From the literature, it appeared that the Stroop effect was more pronounced with this block presentation of pictures (in humans Bar-Haim et al., 2007; McKenna & Sharma, 2004 and in chimpanzees, P. troglodytes, Allritz, Call & Borkenau, 2016). The distractors were displayed at the centre of the screen at the same time as the regular target. The distractors were pictures of 16 × 18.5 cm with matching contrast and luminosity (function ‘Match colour’ in Adobe Photoshop CS6). The category “Object” included a leather ball, a leather bag, a brown stone and a wooden log. The conspecific pictures were chosen to be as realistic as possible, depicting a frontal view of the face and the torso of four unknown adult rhesus macaques. The “Neutral” conspecific included four pictures of individuals with a neutral facial expression. The “Threatening” conspecific included four pictures showing a “open mouth threat” facial expression, frequently displayed by rhesus macaques (Bethell et al., 2016; Hinde & Rowell, 1962). Threatening stimuli (as well as positive stimuli see Hopper et al., 2021) have been shown to have an important distracting effect in macaques (Bethell et al., 2016; Landman et al., 2014). The subjects were not rewarded to touch the distractors and the screen remained the same until the target on the side was touched. During a pilot study (N = 4 subjects, these data were not included in our analysis), we observed that the subjects were, for a long duration, intensely reacting to the pictures of their conspecifics’ faces (lip-smacking, stares and threats) so we set up a maximum response time of 35 s (at the condition that the subject kept looking at the screen). This time period allowed the subject to display a behavioural response, overcome it, and continue the task. If the subject did not touch the target within 35 s, the response latency was not taken into account in the analysis. Each block and trial were counterbalanced across subjects (see Fig. 1). Three sessions of 36 trials were repeated at time point 1 and time point 2 (average days between the time points M ± SD = 12.91 ± 2.84, N = 21; 216 trials total per individual). One male rhesus macaque was not willing to participate in further testing after this task.

Inhibition of action: go/no-go task

In the Go/No-go task the subjects need to respond to frequently presented Go stimuli while withholding a prepotent response to infrequently presented No-go stimuli (Dillon & Pizzagalli, 2007; Duckworth & Kern, 2011).

Design

The apparatus and the general procedure were identical to the Distraction task. A “Go” (red rectangle of 16 × 18 cm) or a “No-go” stimulus (green circle of 16 × 16 cm) appeared randomly in the centre of the screen. The “Go” stimulus was preceded by a high-pitched sound (0.6 s before the stimulus appeared) and the “NoGo” stimulus a low pitch sound to help the subjects anticipate the next trial. The “Go” stimuli appeared 75% of the 40 trials to elicit a prepotent response toward the screen. If the screen was touched outside the stimulus no sound was produced and the trial continued. The “Go” stimulus stayed on the screen until it was touched. We set up a maximum response time (i.e., touching the “Go” stimuli) of 15 s after this the red cross appeared on the screen and the response latency was not recorded. From a pilot study conducted with a shorter response limit, we observed that the subjects frequently left the testing session as they were not rewarded on each “Go” trial. The “No-go” stimulus disappeared if not touched during 2,000 ms and the subject received a reward. If the “No-go” stimulus was touched during this lapse of time, a blank white background appeared for 3,000 ms (as a time out), an “incorrect” sound (with frequency 800, 1,300, 2,000 Hz) was produced and the reward was not given. At first, we fixed a success criterion for the subject’s performances at 80% of correct trials per session, but four macaques never reached this criterion. The performance was therefore measured after five sessions of 40 trials per time point (200 trials in total) for each monkey. These sessions were repeated at time point 2 (average days between the time points M ± SD = 11.95 ± 2.10, N = 20, see Fig. 1).

Inhibition of a cognitive set: reversal learning task

In the Reversal learning the subjects first learn a stimulus-reward contingency. Once a pre-specified criteria is reached this first association is reversed. Subjects must then inhibit a prepotent response to previously correct stimuli and shift responses to a new stimulus-reward contingency (as in Bond, Kamil & Balda, 2007; Tapp et al., 2003). We expected that the subjects would be able to learn a simple discriminant rule and successfully inhibit it.

Design

At the beginning of the task, two stimuli, a “Go” rewarded stimulus (a red square of 15.34 × 15.34 cm) and a “No-go” unrewarded stimulus (a green circle of 15.34 cm of diameter), were displayed at the same time on the screen at counterbalanced locations (left or right of the screen). When the subject touched the “Go” stimulus, the usual “correct” sound was played, the subject received a reward, and a new trial began. If the subject touched the incorrect stimulus the “incorrect” sound was played, the subject did not receive a reward and the two stimuli stayed on the screen until the correct stimulus was touched. If the background was touched nothing happened. We set up a maximum response time (i.e., touching the “Go” stimuli) of 15 s after this the red cross appeared on the screen and the response variables were not recorded. We set up this response limit to keep the subjects engaged with the task. From a pilot study we observed that this period of time allowed the subject to frequently receive a reward and to keep engaged with the task. A session consisted of 40 trials. Once a criterion of success was achieved (75% of correct trials out of 20 trials, i.e., the subjects touched the correct stimulus from the first attempt), the rule was reversed: the correct stimulus became the incorrect and the incorrect the correct. One male macaque was excluded from the study as he did not reach the first criterion. The reversed session was continued until the success criterion was reached again (75% of success for the whole session). Three sessions of the reversed paradigm were repeated at time point 2 (average days between the time points M ± SD = 12.74 ± 5.06, N = 20). One male subject could not participate in the second time point test as it was permanently removed from the facility.

Content validity

Analysis

To validate our battery of inhibitory control tasks our first aim was to assess the content validity of these tasks, for this we wanted to demonstrate that a prepotent response had indeed been triggered (Völter et al., 2018). In the Distraction task, we expected a prepotent response to be triggered by the pictorial distractors, which would increase the response latency of the subjects in a trial with pictures. We expected a greater response latency when a picture was presented, particularly in the trials with the negative stimuli, compared to control trials with no picture (as in Allritz, Call & Borkenau, 2016; Bethell et al., 2016, in non-human primates). We also expected to demonstrate that the subjects would still be able to perform the task by overriding their prepotent response to the distractors, keeping a general high rate of success in the task (i.e., successfully touching the target within the time limit).

To assess the content validity of the Go/No-go task, we wanted to demonstrate that a prepotent response was triggered (an incorrect impulsive action toward the No-go stimulus). Therefore, we investigated the difference in response accuracy in the Go and No-go trials. We expected it to be lower on a No-go trial compared to a Go trial. We also expected the subjects to try to override their impulsive response in the No-go trial, with a greater response latency compared to a Go trial.

In the Reversal learning task, we expected an interference from the previously learnt rule while learning the new rule. We expected a lower probability of success in learning the reversed rule (Rule 2) compared to the first acquisition rule (Rule 1). However, we still expected that the subjects override this interference from the first rule with an overall high accuracy for the second rule. We used a Wilcoxon one sample test to check that the previous task did not reinforce the subjects’ responses toward the red stimuli (even though the location and size were different).

All analyses were conducted in the R environment for statistical computing v.3.6.0 (R Core Team, 2019). We used linear mixed models and general linear model using the functions ‘lme’ from the R package ‘nlme’ v3.1-144 to analyse continuous variable (Harrison et al., 2018) and the function ‘glmer’ from the R package ‘lme4’ v1.1-21 to analyse the binary variable (Bates et al., 2015). The dependent variables were the response latency to touch the target on a trial in the Distraction task and in the Go/No-go task (continuous). As the response latency data presented a skewed distribution, they were log transformed to meet assumptions of normality (Cauchoix et al., 2018; Fazio, 1990; Harald Baayen & Milin, 2010). We excluded latencies below 200 ms (time needed for stimulus perception and motor responses to occur, Harald Baayen & Milin, 2010; Whelan, 2008) and above the time limit. We also recorded the successful completion of a trial in the Go/No-go task and in the Reversal learning task (binary). For all tasks, the random factor included the individual’s identity. We controlled for known influencing factors of inhibitory control by including the following explanatory variables into the models: the sex of the subject (Paul, Harding & Mendl, 2005; Sass et al., 2010), the age (in year, Tapp et al., 2003; Bray, MacLean & Hare, 2014), the experience with pictures (for the Distraction task) and the rank of the subject (Johnson-Ulrich & Holekamp, 2020). To increase the power of the analysis we merged middle ranked individuals and low ranked individuals to have two categories of ranks : either low or high. We had a total of 16 individuals from the higher rank and five from the lower rank. We also controlled for the following task influencing factors: trial, session, time point and type of stimulus (type of picture, no picture vs any type of picture for the Distraction task, type of stimulus: Go vs No-go in the Go/No-go task and type of rule: acquisition or reversed rule in the Reversal learning task). The full model contained all probable explanatory variables (demographic and task determinants). Terms were sequentially dropped from the full model, until the best fitted model contained only those terms that could not be removed without significantly reducing explanatory power (Bates et al., 2015; Harrison et al., 2018). We used the function ‘anova’ from the R package ‘car’ v3.0-6 to compare each model by likelihood ratio test (given as : χ2 (Degree of freedom, N = sample size)). Our significant threshold was p < 0.05. Visual inspection of residual plots did not reveal any obvious deviations from normality. We presented the mean ± the standard error for each effect of the explanatory variable on the outcome. Data are available on an online repository (see Supplemental Materials, DATA SET).

Results

Content validity, distraction task

There were no significant differences between each type of picture (‘Threat’, ‘Neutral’ or ‘Object’) in the overall performances but there was a significant difference in the mean response latency between the trials without pictures (control trial, M ± SE = 3,961.78 ± 104.52 ms, N = 21) and with any type of picture (test trials, M ± SE = 4,496.52 ± 111.17 ms, N = 21, likelihood-ratio test, χ2 (1, N = 21) = 9.98, p = 0.002; see Fig. 2 and Table S1 for more details and for the presentation of the effects of the other explanatory variables). This indicates that the response latency was higher when a picture was present, showing that the subjects were distracted. Nonetheless, when a picture was present the percentage of success on the task (to touch the target within the time limit) was still high (96%) showing that this prepotent response was overridden by the subjects. Thus, the interference from a distractor was inhibited in order to successfully complete the task, these results indicate the content validity of this first task.

Figure 2 Content validity of the inhibitory control tasks.

Distraction task: (A) response latency the absence (With no picture) or in the presence of a picture (With picture); Go/No-go task: (B) proportion of success or (C) response latency in Go or No-go trials; Reversal learning task: (D) proportion of success when first (Rule 1) or second rule learnt (Rule 2). ***indicates that p-value < 0.001.

Content validity, Go/No-go task

There was a significant difference in the mean successful completion of the task between the trials with the Go signal or the No-go signal (likelihood-ratio test, χ2 (1, N = 20) = 3335.6, p < 0.001, see Fig. 2 and Table S2). With only a mean of 47.3 ± 1.09% (N = 20) of correct answers when a No-go signal was presented (i.e., not touching the No-go stimulus), compared to a mean of success (i.e., touch the Go stimulus within the time limit) of 99.8 ± 0.04% (N = 20, see Fig. 2) when it was a Go stimulus presented. This result is showing that a prepotent response, touching the target impulsively was triggered as the subjects, most of the time, made a mistake and touched the unrewarded No-go target associated with a time out. However, there was still a sign that the subjects tried to override their prepotent response, the mean response latency on a No-go trial (M ± SE = 3,123.13 ± 36.68 ms, N = 20) was significantly longer than the one in a Go trial (M ± SE = 2,881.81 ± 91.69 ms, N = 20, see Fig. 2, from the log-transformation of the response latency, likelihood-ratio test, χ2 (1, N = 20) = 600.73, p < 0.001, see Table S3). Please refer to Tables S2 and S3 for the effect of the other explanatory variables on the model. These results indicate that the subject had a prepotent response to touch any time of stimulus, but still tried to overcome this dominant response by slowing down their action toward the screen, this task is thus assessing the inhibition of a prepotent action.

Content validity, reversal learning task

First, we found that the previous task did not influence the subjects’ responses as they did not choose the red stimulus for the first trial above the chance level (chance level = 0.5, mean proportion = 0.53, p = 0.84). There was a significant difference in the mean success in a trial if the rule was the first acquisition or the reversed rule (likelihood-ratio test, χ2 (1, N = 20) = 5.10, p = 0.02, see Table S4 for more details on the other explanatory variables), the probability of success was significantly a little higher in the first rule (65 ± 0.97% of success, N = 20), compared to the reversed rule (63 ± 0.78% of success, N = 20, see Fig. 2). These results indicate that the acquisition rule was interfering with the learning of the reversed rule, despite this interaction 100% of the remaining subject still managed to pass the 75% criterion of success for the second rule (see Table 1 for a summary of the main findings).

Table 1 Summary table presenting the main findings of the study. All tasks have demonstrated content validity, whereby a prepotent response was elicited and overridden. All individual’s performances were repeatable over two time points. The individual’s performances were consistent between the Distraction task and the Go/No-go task but there was no consistency with the Reversal learning task.

	Content validity:
-prepotent response
-overridden	Repeatability of performances over 2 time points	Contextual repeatability	
Distraction task	-Response latency longer when pictures presented
-But still high success	Moderate (R = 0.282)
Adjusted for sex, session and time point (Radj = 0.128)	With the Go/No-go task	
Go/No-go task	-Success on No-go trials lower than Go trials
-Response latency longer for No-go trials	High (R = 0.338)
Not adjusted	With the Distraction task	
Reversal learning task	-Probability of success lower for the reversed than for the acquisition rule
-But still high success	High (R = 0.944)
Not adjusted	With no other tasks	

Temporal repeatability

Analysis

We wanted to assess the temporal repeatability (also known as test-retest reliability) of the individual’s inhibitory performances. For this, we computed, from the response variables (response latency and successful completion of the tasks), scores of inhibitory control. For the Distraction task we computed a Distraction control score, which is the standard method in Stroop task paradigms (as in Allritz, Call & Borkenau, 2016; Bethell et al., 2016, 2019). Distraction control score represented, for each trial, the difference between the mean response latency in all trials without pictures minus the response latency in each trial. For a trial with a picture, a higher score would indicate better control of a Distraction (a shorter response latency). For the second task, the Go/No-go task, to quantify the individual’s ability to inhibit its prepotent action, we calculated the Action control score as the mean percentage of trials when a No-Go was present, and the individual didn’t touch it for each session (Verbruggen & Logan, 2008). We took the last three sessions of each time point of each animal to have a comparable number of sessions between tasks. A higher score would indicate an individual is better at inhibiting the action. Finally, we calculated a Rule control score (as in Tapp et al., 2003), as the difference between the number of trials to reach the criterion of success for the first rule (75% of correct trials for the whole session) minus the number of trials to reach the same criterion for the reversed rule. A higher score would indicate that an individual is better at inhibiting a previous rule when learning a new one.

Once we had these inhibitory control scores, we wanted to assess the temporal repeatability (also known as test-retest reliability) of the individual’s inhibitory control scores, i.e., if the rank of the performances of each subjects within the group were consistent over the two time points (as done in the meta-analysis of Cauchoix et al., 2018). We used the repeatability estimates (R) or Intraclass Correlation Coefficient (ICC) which indicates the amount of variation explained by inter-individual variation of performances in the tasks relative to intra-individual variation (developed by Nakagawa & Schielzeth, 2010). This estimate accounts for both consistency of performances from test to retest (within-subject change), as well as change in average performance of participants as a group over time (Vaz et al., 2013). This test thus assesses the repeatability of the rank of the subjects’ performances within the group between the test and the retest and between contexts. We used the function ‘rpt’, from the ‘rptR’ package v.0.9.22 in R (Nakagawa & Schielzeth, 2010; Stoffel, Nakagawa & Schielzeth, 2017; Vaz et al., 2013). We applied a restricted maximum likelihood function (with 1,000 bootstrapping and 1,000 permutations) and the individual identity was specified as a random intercept effect. The appropriate type of data distribution was adjusted in each model depending on the dependent variable under investigation (“Gaussian” for continuous variables, “Binomial” for binomial variables). We checked Gaussian models for normal distribution of the residuals using the function ‘qqnorm’ from the R package ‘car’ v3.0-6. An individual’s performance was considered as repeatable if the p-value from the Likelihood-ratio test was <0.05. The decision for the qualification of the R estimates, either low, moderate, or high, was based on the work of (Cauchoix et al., 2018). In this meta-analysis, regrouping 44 studies across 25 animal species, the authors computed the repeatability of individual cognitive performances. They found out a mean estimate for the temporal repeatability unadjusted of R = 0.18, 95% CI [0.09–0.28], the R adjusted for test order and individual determinants was Radj = 0.15, 95% CI [0.09–0.21]. We considered that if R ≤ 0.1 the repeatability estimate is low, for 0.1 < R ≤ 0.3, it is moderate and for R > 0.3 it is high.

Once the (R) estimates were calculated, we also needed to consider the influence of confounding factors on the temporal repeatability estimates. For this we calculated adjusted estimates (estimates that adjust for confounding effects which remove fixed effect variance from the estimate, see Nakagawa & Schielzeth, 2010; Stoffel, Nakagawa & Schielzeth, 2017; as done in Cauchoix et al., 2018). Adjusted repeatability can be interpreted as the repeatability given that the level of the confounding factor is known (Nakagawa & Schielzeth, 2010). To calculate these adjusted estimates, we first needed to determine which factors had an effect on the individual’s performances by fitting linear mixed models (LMM) and general linear mixed models (GLMM) (we used the same packages as for the content validity). The outcome variables were either the Distraction control score, the Action control score or the Rule control score. As before, we control for two types of confounding factors: individual determinants (age, sex, rank of the individuals, experience with pictures for the Distraction task), and test determinants (type of stimulus, time point, session and trial). Individual’s identity was included as a random factor. The full model contained all probable explanatory variables (demographic and task determinants). We used the same model selection as before (see content validity) and the function ‘anova’ from the R package ‘car’ v3.0-6 to compare each model and our significant threshold was p < 0.05. Then, we calculated the adjusted repeatability, Radj, by including confounding effects (identified by comparing GLMMs) into the repeatability function. Uncertainty in estimators was quantified by parametric bootstrapping and significance testing was implemented by likelihood ratio tests with a significance threshold of p < 0.05 (Stoffel, Nakagawa & Schielzeth, 2010). We reported 95% confidence intervals (CIs) for parameter estimates based on 1,000 bootstrapping and 1,000 permutations. We reported the result for the adjusted repeatability from the “Link scale approximation”.

Results

Temporal repeatability, distraction task

For the Distraction task, the Distraction control score per individual was moderately repeatable between sessions and the two time points (R = 0.282 ± 0.095, 95% CI [0.093–0.462], p < 0.001). When testing for explanatory factors using LMMs, the model with the variables session, time-point and sex as fixed terms best explained the Distraction control score of the subjects (likelihood-ratio test comparing the best fitted model with the null model: χ2 (5, N = 21) = 119.61, p < 0.001, see Table S5). Males had a lower Distraction control score (likelihood-ratio test, χ2 (1, N = 21) = 9.38, p = 0.002), and this score increased as the number of the session increased (likelihood-ratio test, χ2 (1, N = 21) = 14.02, p < 0.001) and at time point 2 (likelihood-ratio test, χ2 (1, N = 21) = 93.14, p < 0.001). When considering these confounding factors, the adjusted repeatability of the Distraction control score was lower than the unadjusted repeatability (Radj ± SE = 0.128 ± 0.048, 95% CI [0.041–0.241], p < 0.001) but still repeatable (see Fig. 3). Thus, in this Distraction task, the rank of the subjects’ performances within the group was repeatable over the two time points.

Figure 3 Temporal repeatability Radj (adjusted only for the Distraction control score) and 95% bootstrapped confidence intervals for inhibitory control scores.

Y-axis presents the adjusted repeatability for each type of inhibitory control measurement: Distraction task (Distraction control score), Go/No-go (Action control score) and Reversal learning (Rule control score).

Temporal repeatability, Go/No-go task

For the Go/No-go task, the Action control score per session per individual was highly repeatable between sessions and the two time points (R = 0.338 ± 0.105, 95% CI [0.120–0.514], p < 0.001). When testing for confounding factors using GLMMs, none of the variables had a significant effect on the model of the Action control score of the subjects so the temporal repeatability estimates were not adjusted (likelihood-ratio test comparing the full model with the null model: χ2 (7, N = 20) = 6.85, p = 0.44, see Table S6 and Fig. 3). Thus, in this Go/No-go task, the rank of the subjects’ performance within the group was repeatable over the two time points.

Temporal repeatability, reversal learning task

For the Reversal learning task, the Rule control score per session per individual was highly repeatable between the sessions and the two time points (R = 0.944 ± 0.033, 95% CI [0.855–0.981], p < 0.001). When testing for confounding factors using GLMMs, none of the explanatory variables had a significant effect on the models so the temporal repeatability estimates were not adjusted (likelihood-ratio test comparing the full model with the null model: χ2 (7, N = 20) = 3.59, p = 0.61, see Table S7 for more details about the other confounding variables). Thus, in this Reversal learning task, the rank of the subjects’ performances, within the group, was repeatable over the two time points.

To summarize, the rank of the individual performances within the group for inhibitory control scores were consistent over time. While the adjusted temporal repeatability estimates were lower compared to the unadjusted ones, they were still repeatable (see Table 1 for a summary of the main findings).

Contextual repeatability between each task

Analysis

Once we had demonstrated the content validity and the temporal repeatability of our measurements, we were able to look at the cross-contextual consistency of the individual’s performances between the different tasks. We estimated contextual repeatability of our tasks by comparing individual performances on different tasks that putatively measure inhibitory control (as done in the meta-analysis of Cauchoix et al., 2018). We thus wanted to look at the consistency between subject’s performance ranks between each pair of tasks. We use the same repeatability test as before to assess the repeatability of the rank of the subjects’ performances within the group between contexts. As before, we used LMMs and GLMMs to look for confounding factors in order to adjust the contextual repeatability when two tasks were analysed. In the models, the inhibitory control score was the dependent variables and one of the four types of tasks, the sex, the age, the rank, the session, and the time point were the independent variables. Each score was centred and scaled using the function ‘scale’ from the package ‘base’, in R v3.6.3 to allow comparison between the scores of different units. As before, the decision for the qualification of the R estimates, was either low (R ≤ 0.1), moderate (0.1 < R ≤ 0.3), or high (R > 0.3).

Results

The contextual repeatability between the Distraction control score and the Action control score (R = 0.166 ± 0.067, 95% CI [0.041–0.305], p < 0.001), between the Distraction control score and the Rule control score (R = 0.212 ± 0.085, 95% CI [0.049–0.38], p < 0.001) and between the Action control score and the Rule control score (R = 0.138 ± 0.077, 95% CI [0.003–0.303], p = 0.01) were moderate (see Table S8 for a summary of the unadjusted contextual repeatability for all the inhibitory control scores).

The adjusted contextual repeatability (Table 2 summarizes all the adjusted contextual repeatability estimates), between the Distraction control score and the Action control score when controlling for session and time point, remained repeatable and moderate (Radj ± SE = 0.17 ± 0.066, 95% CI [0.045–0.306], p < 0.001, see Table S9). However, between the Action control score and the Rule control score when controlling for sex, the performances between the tasks were not repeatable anymore. (Radj ± SE = 0.101 ± 0.066, 95% CI [0.000–0.24], p = 0.07, see Table S10). Similarly, between Distraction control score and the Rule control score when controlling for sex (Radj ± SE = 0.09 ± 0.058, 95% CI [0.000–0.22], p = 0.07, see Table S11) the performances between the tasks were not repeatable after adjustments. The other scores did not have any confounding variables, so the contextual estimates were not adjusted.

Table 2 Contextual adjusted repeatability estimates of the scores of inhibitory control. Distraction control score (Distraction task), Action control score (Go/No-go task), Rule control score (Reversal Learning task).

Contextual Radj for the scores	Distraction control	Action control	Rule control	
Distraction control	1	–	–	
Action control	✓ 0.170 (p < 0.001)***	1	–	
Rule control	0.09
(p = 0.07)	0.101
(p = 0.07)	1	
Note:

The check mark Indicates that the individual’s performances are significantly repeatable between tasks *** p< 0.0011.

To summarize, we found, for the inhibitory control scores, when adjusted for confounding factors, that the contextual repeatability was only significant between the Distraction task and the Go/No-go (see Table 1 for a summary of the main findings).

Discussion

The aim of this study was to first validate a battery of inhibitory control in non-human primates by assessing the content validity and temporal repeatability of three tasks covering the main domains of inhibitory control. Then, using this battery of tasks, we wanted to investigate the structure of inhibitory control by looking at the contextual consistency of subjects’ performances between these tasks. First, we found a response pattern characteristic of inhibitory control in each of the three tasks, an indicator of content validity. A prepotent response (an interference from a pictorial distractor, a dominant motoric response, and a pre-learned rule) was inhibited by the subjects to successfully achieve the goal of the task. We then confirmed that the performances of the subjects were repeatable across 2 time points, thus validating the test-retest reliability of our tasks. Finally, our results gave an insight of the structure of inhibitory control by demonstrating that the individual performances between the Distraction task and the Go/No-go tasks, even after adjustments for confounding factors (session and time point), were also consistent, indicating that these tasks seem to capture the same cognitive process. However, after adjustment, the individual performances between the Reversal learning task and the other tasks of inhibitory control were not consistent, interestingly indicating the absence of a common underlying ability (see Table 1 for a summary of the main findings).

The first step in validating a battery of inhibitory control measurement was to demonstrate their content validity. Looking closely at the pattern of response, our results indicated that a prepotent response was generated from the test conditions. In other research studies, using the Reversal learning task and motor inhibition tasks, in dogs (C. familiaris, Bray, MacLean & Hare, 2014; Brucks et al., 2017; Marshall-Pescini, Virányi & Range, 2015; Vernouillet et al., 2018), wolves (Marshall-Pescini, Virányi & Range, 2015) or in pheasants (Phasianus colchicus, van Horik et al., 2018, 2019), the tasks were similarly producing a prepotent response. The same pattern of response was found in the Distraction tasks in non-human primates (Allritz, Call & Borkenau, 2016; Bethell et al., 2016; Landman et al., 2014). Unfortunately, this inhibitory pattern of response is rarely systematically investigated. For example, the cylinder task has been used in a large comparative study, of more than 36 species, to draw conclusions about the evolution of inhibitory control (MacLean et al., 2014). However, the content validity of this task has been recently challenged (Kabadayi, Bobrowicz & Osvath, 2018; Shaw & Schmelz, 2017). Thus, to first demonstrate the content validity of a cognitive task seems a crucial step in order to both justify its use and draw evolutive conclusions from it.

Another critical step was to make sure that these tasks were repeatable over time. A lack of temporal repeatability can be detrimental in subsequent assessment of validity (Friedman & Miyake, 2017; Paap & Oliver, 2016). As expected, we found moderate and significant, temporal estimates. Our mean estimate of R = 0.40 is higher than mean estimates found in common cognitive tasks in animals (from the meta-analysis of Cauchoix et al., 2018, the mean estimate is R = 0.18). In the animal cognition literature, inhibitory control temporal estimates are ranging from very low (R = 0.012) to very high values (R = 0.975) (Ashton et al., 2018; Cauchoix et al., 2017, 2018); our range of estimates is similarly diverse (from R = 0.128 to 0.944) but still significant as expected. The confirmation of the temporal repeatability of any cognitive measurements seems also a crucial step before establishing the validity of any tasks.

Once we established the content validity and temporal repeatability of our measurements, we evaluated the cross-contextual consistency of the inhibitory control tasks. We first found that all the unadjusted estimates were significant. These results were similar to the ones obtained in Australian magpies which obtained a Spearman rank order correlation estimate of r = 0.433 between the cylinder task and the reversal learning task (Ashton et al., 2018). Similarly, in a large interspecies study, MacLean et al. (2014), found that performance on the A not B and cylinder task was strongly correlated (r = 0.53). However, using the data from the study of MacLean et al. (2014), looking from an individual difference perspective, Völter et al. (2018), did not find any correlation between the inhibitory control tasks. When controlling for confounding factors, the adjusted repeatability between the Distraction task and the Go/No-go task were lower but still significant, this could indicate a common underlying ability. The decrease in the value of the adjusted contextual estimates were similar to the one found in Cauchoix et al. (2018). This could be because the confounding factors, that vary between individuals, reduce the between-group variance and thus the repeatability (Nakagawa & Schielzeth, 2010). These results confirm the importance of controlling for confounding factors when assessing contextual repeatability.

Surprisingly, the adjusted contextuality estimates between the Reversal learning task and the two other tasks of inhibitory control were not significant. It seems that the Distraction tasks (inhibition of a distraction) and the Go/No-go (inhibition of an impulsive action), share a common underlying inhibitory ability, i.e., to inhibit an impulsive, unconscious response to a stimulus. However, between the Reversal learning task (inhibition of a cognitive set) and the other tasks no clear pattern emerges that would support the notion of a common cognitive ability. These results reproduce the same pattern found in human research, with correlation between two inhibition-related functions: the “resistance to distractor interference” (similar to our definition of Distraction inhibition) and “prepotent resistance interference” (similar to our definition of action inhibition) but not with “resistance to proactive interference” (similar to our definition of cognitive set inhibition; Friedman & Miyake, 2004). Interestingly, similar results were also demonstrated in several studies in canids, specifically designed to understand the structure of inhibitory control. Authors found a lack of correlation between the detour task or cylinder task (inhibition of an impulsive action) and the A-not-B task (inhibition of a cognitive set; Bray, MacLean & Hare, 2014; Brucks et al., 2017; Fagnani et al., 2016; Marshall-Pescini, Virányi & Range, 2015; Vernouillet et al., 2018). Thus, the inhibition of an external distractor and the inhibition of a prepotent motor response seem to share the same underlying inhibitory ability but not the inhibition of a previously learned rule. It is possible that the inhibition of an impulsive and prepotent, stimulus driven response, relies more on cognitively low demanding construct, such as a simple bottom-up inhibitory control function (Nigg, 2017). On the contrary, in the learning of a new rule, it could be required, in addition to inhibitory control, to employ a higher deliberate cognitive ability, a top-down function, relying on mental representations (e.g., working memory or set shifting, Dillon & Pizzagalli, 2007; Nigg, 2017). These results are at odds with the hypothesis that all three inhibition-related functions are measuring some common ability. Thus, our results provide a new insight in favour of the multifaceted structure of inhibitory control in a non-human primate. It would be difficult to broaden our interpretation to other species because of the potential species-specific differences in the nature of inhibitory control. Inhibitory control could be a general construct in some taxa but a family of independent components in other taxa.

Understanding the structure of inhibitory control is particularly crucial as impairments in inhibitory control have been associated with several psychopathologies. For instance, in children suffering from ADHD, it is still unclear if it is inhibitory control as a general ability which is impaired or only some independent components (Gaultney et al., 1999; Nigg, 2000). Consistent with this last view, there is some evidence that individuals with ADHD are impaired on tasks measuring response inhibition, whereas it remains unclear if these patients are also impaired in cognitive inhibition tasks (Gaultney et al., 1999; Nigg, 2000). However, we need to be cautious in the comparison of results between studies using different types of contextual validity analysis. On one hand, some studies used correlations analysis (Bray, MacLean & Hare, 2014; Duckworth & Kern, 2011; Fagnani et al., 2016; Vernouillet et al., 2018); which focuses on the strength of association between two means of performances (Liu et al., 2016; McGraw & Wong, 1996). On the other hand, some authors used repeatability estimates or intraclass correlation based on variance analysis (Ashton et al., 2018; Cauchoix et al., 2018; Shaw, 2017). Unlike correlation, in this agreement analysis, the emphasis is on the degree of concordance between individual performances (Koo & Li, 2016; Liu et al., 2016; McGraw & Wong, 1996). Furthermore, repeatability analysis, unlike correlation, allowed researchers to control for confounding factors (such as individual or temporal determinants, Nakagawa & Schielzeth, 2010; Vaz et al., 2013). It is thus possible to have two sets of scores that are highly correlated, but not repeatable (Zaki et al., 2013). We can thus look for patterns between results of different studies, but we should be careful in making stronger assertions.

Moreover, the interpretation of the repeatability estimates must be drawn carefully because it assumes that inhibitory control performances on each cognitive test is independent of other idiosyncratic task demands (e.g., learning, problem solving), or individual characteristics (e.g., motivation, personality traits; see Cauchoix et al., 2018; Griffin, Guillette & Healy, 2015; Kabadayi, Bobrowicz & Osvath, 2018). For example, motivation to get a reward is an important confounding factor. If the reward is visible it strongly affects the subject’s ability to inhibit its response (Brucks et al., 2017; Kabadayi, Bobrowicz & Osvath, 2018). As in the inhibitory control field, this task impurity problem is particularly strong, we should be careful to label any common factor inhibitory control (or a suite of inhibitory control abilities) as it remains unclear what the shared variance represents (Friedman & Miyake, 2004; Völter et al., 2018). These results are in line with the claim that the nature of inhibitory control is not unitary but are more likely a collection of sub-components intertwined with other cognitive processes that may or may not be engaged on specific contexts that require inhibition (Beran, 2018; Diamond, 2013; Dillon & Pizzagalli, 2007; Friedman & Miyake, 2004; Nigg, 2017).

To minimize this task impurity problem, a measurement of another influencing construct, e.g., a memory task (known to tap mostly in memory ability) should ideally be incorporated in a battery of inhibitory control tasks. In this way, researchers could compare the performances in the inhibitory control tasks and the memory task to try to disentangle the different constructs involved. In addition, we could incorporate measures of behaviours characteristic of other involved constructs (Cauchoix et al., 2018). For example, an eye tracker could be used to record the rate of gaze switching direction between stimuli to control for attention. Similarly, the occurrence of facial expression or body scratches could be used to assess the emotional arousal or stress of the subjects. Once the effect of confounding factors is clearer, researchers could focus on a broader approach of inhibitory control looking at the factors influencing its evolution, such as species’ social life or ecology.

Another limitation we faced, common when working with primates, is the low sample size which might decrease the power of our analysis (Koo & Li, 2016; Paap & Oliver, 2016; Völter et al., 2018). Moreover, our results are potentially only representative of one sample of one population of captive rhesus macaques. We hope that this study will be replicated on a larger sample size, using for instance, a large-scale collaborative project across laboratories or field sites (e.g., Primates et al., 2019).

Conclusion

To summarize, we have developed a battery of touchscreen tasks of inhibitory control which demonstrated content validity and temporal repeatability. We showed a consistency of performance between the inhibition of a distraction and the inhibition of an action, representing a response-driven basic form of inhibition, this was not found for inhibition of a pre-learned rule. This task battery provides us with a new insight into the structure of inhibitory control in a non-human primate. This ability seems to be composed of intertwined sub-processes, which might or might not rely on other cognitive constructs. Inhibitory control could be divided in sub-components, with on one hand, a cognitively low demanding process involving the inhibition of a prepotent, stimulus driven response and on another hand a more controlled, deliberate inhibition of a mental state. It seems crucial that future studies focus on a better understanding of this ability given the importance of inhibition-related processes in successful day-to-day living.

Supplemental Information

Supplemental Information 1 Results of LMMs for the log transformation of the response latency in the Distraction task.

Confounding factors were divided in individual (sex, age, rank and experience with picture) and experimental determinants (session and time point). All full models included the individual ID as a random factor. The Estimates (representing the change in the dependent variable relative to the baseline category of each predictor variable), Standard Error, t-value and p-value using maximum likelihood method. The variables in bold stimulus, age, trial and time point had a significant effect on the models. 4,094 data points were analysed.

Note. Number of subjects 21 Likelihood-ratio test comparing the best fitted model (with session, time point and age as explanatory variables) with the null model : χ2 4 = 296.02, p < 0.0001. The subjects had a longer response latency as they get older (χ2 1 = 9.086, p < 0.01), and their response latency were shorter as session (χ2 1 = 4.798, p < 0.05) and time point (χ2 4 = 276.165, p < 0.0001) increased.

Click here for additional data file.

Supplemental Information 2 Results of GLMMs for the success in the Go/No-go task.

Confounding factors were divided in individual (sex, age and rank) and experimental determinants (session and time point). All full models included the individual ID as a random factor. The Estimates (representing the change in the dependent variable relative to the baseline category of each predictor variable), Standard Error, z-value and p-value using maximum likelihood method. The type of stimulus (Go or No-go) and session had a significant effect on the models. 7,783 data points were analysed.

Note. Number of subjects 20 Likelihood-ratio test comparing the best fitted model (with type of stimulus and session as explanatory variables) with the null model: χ2 2 = 3335.6, p < 0.0001. The success on a trial was higher as the number of the session increased: χ2 1 = 6.172, p < 0.05.

Click here for additional data file.

Supplemental Information 3 Results of LMMs for the log transformation of the response latency in the Go/No-go task.

Confounding factors were divided in individual (sex, age and rank) and experimental determinants (session and time point). All full models included the individual ID as a random factor. The Estimates (representing the change in the dependent variable relative to the baseline category of each predictor variable), Standard Error, t-value and p-value using maximum likelihood method. Only the variable in bold stimulus had a significant effect on the models. 7,783 data points were analysed.

Note. Number of subjects 20 Likelihood-ratio test comparing the best fitted model (with type of stimulus as explanatory variables) with the null model: χ2 1 = 600.73, p < 0.001.

Click here for additional data file.

Supplemental Information 4 Results of GLMMs for the success in the Reversal learning task.

Confounding factors were divided in individual (sex, age and rank) and experimental determinants (session and time point). All full models included the individual ID as a random factor. The Estimates (representing the change in the dependent variable relative to the baseline category of each predictor variable), Standard Error, z-value and p-value using maximum likelihood method. The variables in bold rule, trial and session had a significant effect on the models. 6,686 data points were analysed.

Note. Number of subjects 19 Likelihood-ratio test comparing the best fitted model with the null model: χ2 4 = 27.74, p < 0.001. The success on a trial was higher as the trials (χ2 1 = 4.101, p < 0.05) and session increased ( χ2 1 = 11.687, p < 0.05 ).

Click here for additional data file.

Supplemental Information 5 Results of LMMs for the Distraction control score (the Distraction task).

Confounding factors were divided in individual (sex, age, rank and experience with picture) and experimental determinants (session and time point). All full models included the individual ID as a random factor. The Estimates (representing the change in the dependent variable relative to the baseline category of each predictor variable), Standard Error, t-value and p-value using maximum likelihood method. Only the variables in bold sex, session and time point had a significant effect on the models. 346 data points were analysed.

Click here for additional data file.

Supplemental Information 6 Results of LMMs for the Action Control Score (Go/No-go).

Confounding factors were divided in individual (sex, age and rank) and experimental determinants (session and time point). All full models included the individual ID as a random factor. The Estimates (representing the change in the dependent variable relative to the baseline category of each predictor variable), Standard Error, t-value and p-value using maximum likelihood method. None of the variables had a significant effect on the models. 120 data points were analysed.

Click here for additional data file.

Supplemental Information 7 Results of GLMMs for the Rule Control Score (Reversal learning task).

Confounding factors were divided in individual (sex, age, rank) and experimental determinants (session and time point). The Estimates (representing the change in the dependent variable relative to the baseline category of each predictor variable), z-value and p-value using maximum likelihood method. None of the variables had a significant effect on the model. 38 data points were analysed.

Click here for additional data file.

Supplemental Information 8 Contextual unadjusted repeatability estimates of the scores of executive function and inhibitory control.

Distraction control score (Distraction task), Action control score (Go/No-go) and Rule control score (Reversal Learning) are represented.

✔ indicates that the individual’s performances are significantly repeatable between tasks. *p < 0.05, **p < 0.01, ***p < 0.001

Click here for additional data file.

Supplemental Information 9 Results of LMMs for the Distraction task (Distraction control scores) and the Go/No-go task (Action Control Score).

Confounding factors were divided in individual (sex, age and rank) and experimental determinants (session and time point). The Estimates (representing the change in the dependent variable relative to the baseline category of each predictor variable t-value and p-value using maximum likelihood method. The variables in bold session and time point had a significant effect on the models. 237 data points were analysed.

Click here for additional data file.

Supplemental Information 10 Results of LMMs for the Go/No-go (Action control scores) and the Reversal learning task (Rule Control Score).

Confounding factors were divided in individual (sex, age and rank) and experimental determinants (session and time point). All full models included the individual ID as a random factor. The Estimates (representing the change in the dependent variable relative to the baseline category of each predictor variable t-value and p-value using maximum likelihood method. Only the variable in bold sex had a significant effect on the model (when comparing with the full model). 158 data points were analysed.

Click here for additional data file.

Supplemental Information 11 Results of LMMs for the Distraction task (Distraction control scores) and the Reversal learning task (Rule Control Score).

Confounding factors were divided in individual (sex, age and rank) and experimental determinants (session and time point). All full models included the individual ID as a random factor. The Estimates (representing the change in the dependent variable relative to the baseline category of each predictor variable), Standard Error, t-value and p-value using maximum likelihood method. The variable in bold had a significant effect on the models. 155 data points were analysed.

Click here for additional data file.

We are grateful to Margot Moniot and Elen Stanton for their help in collecting the data. Thank you to Florent Le Moël and Alexandre Montlibert for their help in creating the MATLAB scripts. Many thanks to Dr. Claire Witham and all the caretakers from the MRC, UK for their help in coordinating in situ the collection of the data. Thank you to Dr. Claire Witham for letting us use her pictures of the macaques.

Additional Information and Declarations

Competing Interests

Author Contributions

Animal Ethics

Field Study Permissions

Data Availability

The authors declare that they have no competing interests.

Louise Loyant conceived and designed the experiments, performed the experiments, analyzed the data, prepared figures and/or tables, authored or reviewed drafts of the paper, and approved the final draft.

Bridget M. Waller conceived and designed the experiments, authored or reviewed drafts of the paper, and approved the final draft.

Jérôme Micheletta conceived and designed the experiments, authored or reviewed drafts of the paper, and approved the final draft.

Marine Joly conceived and designed the experiments, authored or reviewed drafts of the paper, and approved the final draft.

The following information was supplied relating to ethical approvals (i.e., approving body and any reference numbers):

The Animal Welfare and Ethical Review Body of the University of Portsmouth provided full approval for this research (AWERB no. 4015B).

The following information was supplied relating to field study approvals (i.e., approving body and any reference numbers):

Field experiments were approved by the Animal Welfare and Ethical Review Body of the Medical Research Council, Center for Macaques (MRC-CFM) in Porton Down, United Kingdom (ARWEB no. CFM2019E00).

The following information was supplied regarding data availability:

The data set, MATLAB codes and stimuli and additional materials (training phases, rank calculations and supplementary results) are available at GitHub: https://github.com/Psychology-inhibitory-control/ARTICLE-VALIDITY.git.

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
