# Peer review of "Validation of a battery of inhibitory control tasks reveals a multifaceted structure in non-human primates"

_PeerJ, doi:10.7717/peerj.12863_

## Round 0.1 · original submission · Major Revisions

Dear Louise

Thank you for your submission, which has now been seen by three reviewers. All are positive, and agree your paper makes a valuable contribution. I've gone for major revisions here, as reviewer 1 does query whether one of your procedures tests what it is claiming to test, and also has some suggestions for improving the structure and clarity of your paper. I have every confidence you can address these concerns, and also deal with the comments of the other reviewers, but I do think the issues raised by reviewer 1 are important and need to be dealt with adequately before reaching a final decision.

All the very best,
Lou

Reviewer 1 ·

Basic reporting

I want to commend the authors on their thorough and careful reporting. I especially appreciated Figure 1, which really helped to orient me as I was reading through the methods and results.

However, the introduction is rather long. I think that some of the descriptions of content validity, repeatability, and impurity could be cut back considerably (ie lines 92-139).

Second, when introducing your study species (ie lines 147-151) I think it would be worth presenting what’s already known about macaques’ performance in inhibitory control tasks, especially those that you include in your study, to help frame and inform your hypotheses (e.g. Hopper et al., 2021; Lauwereyns et al 2000; Rathke & Fischer, 2020; Reeves et al. 2017; Vardanjani et al., 2021)

Lastly, I found it a little confusing that the analyses description was provided before the detailed task descriptions. I suggest reordering this.

Cited References
Hopper et al 2021. A comparative perspective on three primate species' responses to a pictorial emotional Stroop task. Animals, 11(3), 588

Lauwereyns et al 2000. Interference from Irrelevant Features on Visual Discrimination by Macaques (Macaca fuscata): A Behavioral Analogue of the Human Stroop Effect. J. Exp. Psychol. Anim. Behav. Proc. 26, 352–357

Rathke and Fischer 2020 Differential ageing trajectories in motivation, inhibitory control and cognitive flexibility in Barbary macaques (Macaca sylvanus). Philosophical Transactions of the Royal Society, 375(1811).

Reeves et al. 2017. Within-session reversal learning in rhesus macaques (Macaca mulatta). Animal Cognition 30, 9750983

Vardanjani et al., 2021 The effects of emotional stimuli and oxytocin on inhibition ability and response execution in macaque monkeys. Behavioural Brain Research, 412, 113409.

Experimental design

In this study, the authors report a series of touchscreen tasks presented to macaques to assess inhibitory control. My main concern with this study is that I am not convinced that the first test is actually testing what is claimed. I suggest that this is presented a “training” exercise (i.e. familiarizing the macaques with touching stim on a touchscreen) and that this could even be moved to supplemental materials so that the focus of the analysis can be on the tasks that appear to actually test relevant metrics. In addition to this more fundamental concern, I have some other questions and request for clarifications that I present in the order in which they arose as I read the article.

For the first task, I am not sure what an incorrect response would be given that there was no choice to be made – the monkeys just had to touch a red triangle. Please clarify this. Relatedly, and more importantly, this task is very simple and represents the basic training phase of any touchscreen training so it seems a leap to describe this as an executive function task. What planning was required by the macaques? Also, if the touchscreens were new to the monkeys at the start of this study, why would you expect repeatability in this? Surely there would be a learning curve with time (as measured in decreasing response latencies)? How familiar with the touchscreens were the macaques prior to this and what other stimuli had you trained them on?

With the second task, what happened if the macaque touched the distractor stimuli? And was the distractor presented simultaneously with the target? This was not clear (line 433).

Line 473 – but now the time limit for the response was longer. Have you analyzed their responses to see how many responses were made within the previous trial time limit?

Line 525 – why was the no-go stim presented for such a brief period? Are the monkeys able to respond (incorrectly) this quickly? How does this relate to their average response latencies in other tasks and how does this method compare to other go/no-go tasks used previously? It just seems that it’s somewhat setting the monkeys up for success without allowing them to fail.

Line 552 “there was still a sign that the subjects tried to override their prepotent response” this seems like a big claim. It could just be that the red triangle was more familiar to the monkeys and so they were more likely to reach for it than the unfamiliar green circle, not because they “knew” they shouldn’t respond to that second stim.

Validity of the findings

I have some concerns regarding the presentation and interpretation of task 1 - see my above comments

Additional comments

Minor comments
Line 185 is the “free roaming room” and indoor space or did these macaques have outdoor access?

Line 203 what was the average group size that the monkeys were housed in?

Line 205 “from the higher rank” this is unclear, please clarify. Did you simply classify the macaques as high vs low ranked from the David’s score and 15 were high and 6 were low?

Line 240 “the timer was paused,” is this a built in timer within the software or were you manually recording the times? This makes it a little unclear.

Line 409 but see Hopper et al. 2021 who only found that positive stim interfered with macaques’ responses in a Stroop task.

Line 465 “wasn’t” should be “was not”

Line 670 could these comparisons be presented in a table? I found it hard to parse all these comparisons from the text and I think a summary table would be more accessible.

Reviewer 2 ·

Basic reporting

no comment

Experimental design

The tasks used are appropriate and overall the methodology is well explained.

Validity of the findings

Due to the nature of the study, the methods and results sections are very long and a bit difficult to follow. The findings are valid and the conclusions drawn by the authors are in line with their findings. This is useful manuscript for animal cognition researchers and I have relatively minor, specific comments for the authors.

Additional comments

Line 21: is inhibitory control only restricted to unproductive responses? Inhibition may also work on responses that are productive, but potentially less so than delayed responses. I suggest you revise this section of the abstract to be more consistent with your definition of inhibition in the introduction, where productivity is better explained.
Line 66: check typo and add comma to 33,000
Line 204: section on dominance rank is vague. What was the minimum number of interactions used to calculate rank for each animal. What do you mean as ‘higher rank’?
Line 293: how many study subjects did you have in each of the three rank categories?
Line 297: since model selection has been criticised by some authors, wouldn’t it be simpler/better to just compare the full model to a null model containing all your control variables but not the test variables? From the way you describe the aims in the introduction, this model selection approach seems unnecessary. Please clarify point this or revise the analyses presented.
Moreover, it is not clear how many data points were used for these models. This is important to understand whether your sample size could ‘accept’ the number of variables you used in the model.
Due to the nature of the study, the results section is very long and for the reader it’s difficult to keep track of the different findings. I suggest you add a summary table with the main findings of your study, if space is available.
Discussion: unless I missed it, you don’t mention anywhere in your manuscript that a possible source of variance in the literature on inhibitory control is due to species-specific differences. For example, it is possible that inhibition could be a general construct in some taxa and a family of multi-components in other taxa. It would be useful to briefly discuss this option in your manuscript, especially in relation to whether differences in support of one of these two hypotheses may be related to the species being tested.

Reviewer 3 ·

Basic reporting

Review of “Validation of a battery of inhibitory control tasks reveals a multifaceted structure in non-human primates (#65913)” submitted to PeerJ.

This article tries to answer whether Inhibitory control (IC) is a general construct or a family of distinct sub-components, by testing monkeys in touchscreen tasks aimed to target 3 aspects of IC (a new battery of tasks, another goal of the current MS). The authors found some support for a general construct but also some support for distinct sub-components. I found the MS easy to read and generally convincing (not least, as the authors place their findings well into the literature and also do not oversell their results).

Was there “Clear, unambiguous, professional English
language used throughout.”? Yes.

Did “Intro & background show context.”? Yes.

Was the “Literature well referenced & relevant.”? Yes.

Were “Figures relevant, high quality, well
labelled & described.”? Yes.

Was the “Raw data supplied”? The authors claim that this was supplied – but I could not find the data (and neither a functioning link to that data in a repository) myself among the files I received.

Experimental design

Was the “Original primary research within Scope of
the journal.”? Yes.

Was the “Research question well defined, relevant
& meaningful.”? Yes.

“It is stated how the research fills an identified knowledge gap.”? Yes.

Was there “Rigorous investigation performed to a
high technical & ethical standard.”? Yes. I especially liked the “distraction of other group members” aspect. Note that I did not see the ethic board decision letters.

Were “Methods described with sufficient detail &
information to replicate.”? Yes.

Validity of the findings

Impact and novelty: not assessed, as per PeerJ design.

Was “All underlying data provided;
and were they robust, statistically sound, &
controlled.”? Yes.

Were “Conclusions well stated, linked to
original research question & limited to
supporting results.”? Yes.

Additional comments

“to first demonstrate the content validity of a cognitive task seems a crucial step in order to both justify its use and draw evolutive conclusions from it.” I could not agree more. It is nice to see the current efforts to overcome such hurdles.
I wish to congratulate the authors on a nice study and nice paper that goes quite some way in this direction.

- The authors mention pilot subjects. Could the authors explain whether this data was entered into the main data set? Could they also state whether these subjects were excluded from later testing?

- Iine 711: word repeat

---

## Round 0.2 · accepted · Accept

Thanks for all your efforts here! All looking good now.